# Toll-like Receptor Response to Hepatitis C Virus Infection: A Recent Overview

**DOI:** 10.3390/ijms23105475

**Published:** 2022-05-13

**Authors:** Mohammad Enamul Hoque Kayesh, Michinori Kohara, Kyoko Tsukiyama-Kohara

**Affiliations:** 1Department of Microbiology and Public Health, Faculty of Animal Science and Veterinary Medicine, Patuakhali Science and Technology University, Barishal 8210, Bangladesh; 2Department of Microbiology and Cell Biology, Tokyo Metropolitan Institute of Medical Science, Tokyo 156-8506, Japan; kohara-mc@igakuken.or.jp; 3Transboundary Animal Diseases Centre, Joint Faculty of Veterinary Medicine, Kagoshima University, Kagoshima 890-0065, Japan

**Keywords:** hepatitis C virus, infection, innate immunity, Toll-like receptor, cytokines

## Abstract

Hepatitis C virus (HCV) infection remains a major global health burden, causing chronic hepatitis, cirrhosis, and hepatocellular carcinoma. Toll-like receptors (TLRs) are evolutionarily conserved pattern recognition receptors that detect pathogen-associated molecular patterns and activate downstream signaling to induce proinflammatory cytokine and chemokine production. An increasing number of studies have suggested the importance of TLR responses in the outcome of HCV infection. However, the exact role of innate immune responses, including TLR response, in controlling chronic HCV infection remains to be established. A proper understanding of the TLR response in HCV infection is essential for devising new therapeutic approaches against HCV infection. In this review, we discuss the progress made in our understanding of the host innate immune response to HCV infection, with a particular focus on the TLR response. In addition, we discuss the mechanisms adopted by HCV to avoid immune surveillance mediated by TLRs.

## 1. Introduction

Hepatitis C virus (HCV) infection is a major global health burden [1,2]. HCV infection frequently causes chronic hepatitis, liver cirrhosis, and hepatocellular carcinoma (HCC) [3]. According to the World Health Organization, 58 million people worldwide are chronically infected with HCV, with approximately 1.5 million new infections occurring each year [4]. HCV is an enveloped, positive-sense, single-stranded RNA virus belonging to the genus *Hepacivirus* and the family *Flaviviridae* [5]. HCV has a ∼10-kb long genome, encoding a large polyprotein of approximately 3000 amino acids that is processed by host and viral proteases into three structural (core, E1, and E2) and seven non-structural (NS) proteins (p7, NS2, NS3, NS4A, NS4B, NS5A, and NS5B) [6]. HCV has high genetic diversity resulting in seven major genotypes and more than 60 subtypes [7].

The innate immune system, an essential component of host immunity, plays a key role in the initial detection of invading pathogens, including viruses, and subsequently activates adaptive immunity, thereby playing an important role in the early control of viral infection [8,9,10,11]. The host innate immune response is activated upon microbial invasion and detection of evolutionarily conserved structures found on pathogens, called pathogen-associated molecular patterns (PAMPs), by germ-line-encoded pattern recognition receptors (PRRs) [12]. PRRs also recognize molecules released by damaged cells, known as damage-associated molecular patterns [13]. Different PRRs, including Toll-like receptors (TLRs), RIG-I-like receptors (RLRs), NOD-like receptors (NLRs), AIM2-like receptors (ALRs), C-type lectin receptors (CLRs), and intracellular DNA sensors such as cGAS, are key innate immune components that recognize viral components such as viral nucleic acids and proteins [14,15]. However, TLRs are the most widely characterized PRRs, constituting key components of innate immunity, and are involved in the early interaction with PAMPs of invading microbes [16].

TLRs are evolutionarily conserved type I transmembrane proteins that contain a conserved structure of an N-terminal ectodomain of leucine-rich repeats, a single transmembrane domain, and a cytosolic Toll/interleukin (IL)-1 receptor (TIR) domain [11,17]. The cytosolic TIR domain is responsible for the activation of downstream signaling, and TLR signaling pathways are regulated by TIR domain-containing cytosolic adaptor proteins such as myeloid differentiation factor 88 (MyD88), MyD88 adaptor-like (MAL or TIRAP), TIR-domain-containing adaptor protein inducing interferon (IFN)-β (TRIF or TICAM1), TRIF-related adaptor molecule (TRAM or TICAM2), and sterile α- and armadillo-motif-containing protein (SARM) [8,18,19]. The adaptor protein MyD88 is used by nearly all TLR signaling pathways, except TLR3 [20]. Members of each TLR family have similar functions across species [21,22]. TLRs are encoded by a large gene family, and different organisms appear to encode a certain number of TLRs. For example, the TLR family comprises 10 members (TLR1–TLR10) in humans and 12 members (TLR1–TLR9 and TLR11–TLR13) in mice [23]. TLRs can be localized either on the cell surface, such as TLRs 1, 2, 4, 5, 6, and 10, or in intracellular compartments (e.g., the endoplasmic reticulum, endosome, lysosome, or endolysosome) such as TLRs 3, 7, 8, and 9 [23,24,25]. TLR1, TLR2, TLR4, TLR5, and TLR6 play pivotal roles in viral protein recognition [26]. To recognize viral double-stranded RNA, single-stranded RNA, and DNA, the membrane proteins TLR3, TLR7/8, and TLR9 are used, respectively [27,28,29,30].

TLR-induced innate immune responses appear to be a prerequisite for the generation of most adaptive immune responses and play a central role in shaping such responses [25,31]. To provide protection against invading microbes, TLR responses ultimately lead to the induction of IFNs, cytokines, and chemokines via several distinct signaling pathways, which is very important in limiting the spread of infection [14,31,32]. TLR agonists appear to be potential immunomodulators for treating infections and play a significant role in modulating immunotherapeutic effects [33,34,35,36]. However, the TLR response may not always be beneficial to the host, but a dysregulated response may lead to immune-mediated pathology rather than protection [37,38,39,40]. Therefore, a proper understanding of the TLR response in any infection, including HCV infection, is critical. Against this background, we discuss here the current progress made in our understanding of the host TLR response in HCV infection and the mechanisms adopted by HCV to avoid TLR-mediated immune surveillance, which may help in devising new therapeutic or preventive strategies.

## 2. TLR Response to HCV Infection

A complex interplay between the IFN system and viral countermeasures exists in HCV infections [41]. During viral replication, HCV PAMPs can be recognized as non-self by PRRs, leading to the activation of innate and adaptive immune responses [42]. TLRs are important PRRs that recognize PAMPs present in HCV [43]. TLRs are key triggering molecules for cytokine production, and TLR signaling pathways provide a link between innate and acquired immunity [15,44]. Different TLRs have been found to interact with HCV proteins and nucleic acids. It has been reported that HCV is sensed by TLR3 through the detection of dsRNA intermediates in infected hepatoma cells, which may activate the TLR3-signaling cascade and lead to the production of type I and II IFNs, expression of interferon-stimulated genes, and proinflammatory cytokines limiting HCV replication [45]. Induction of TLR2 and TLR4 expression and modulation of the proinflammatory response by HCV proteins have been reported in Raji cells and peripheral blood mononuclear cells (PBMCs) [46,47]. Several other studies also found increased expression of TLR2 and TLR4 mRNA in chronic hepatitis patients compared to controls [48,49]. However, TLR4 mRNA expression was downregulated in cirrhotic patients when compared to chronic hepatitis patients [48]. Hypo-responsiveness to TLR ligands has been reported in patients with chronic HCV infection [50]. Differential expression of TLRs has been reported in HCV-infected patients [51,52], where increased expression of tumor necrosis factor (TNF)-α, IL-6, and IL-12 p35 in PBMCs was also reported [52]. It has been observed that NS5A can activate the promoter of the TLR4 gene in both hepatocytes and B-cells, and enhanced TLR4 expression induces the induction of IFN-β and IL-6 production in human B-cells [46].

The HCV core and NS3 proteins play significant roles in HCV pathogenesis; it has been shown that they can trigger proinflammatory cytokine production in monocytes, inhibit myeloid dendritic cell accessory cell functions, and provide immunoinhibitory effects via IL-10 induction [53]. Other studies have reported that HCV viral proteins, including HCV core and NS3, could be recognized by TLR2, triggering the activation of inflammatory cells [54,55]. Impaired recognition of HCV core and NS3 proteins caused by the R753Q SNP in TLR2 has been reported [55]. HCV core and NS3 proteins may enhance the activity of IL-1 receptor-associated kinase (IRAK), phosphorylation of p38, extracellular regulated (ERK), and c-jun N-terminal (JNK) kinases and induce the activation of activator protein 1 (AP-1) [54]. In an in vitro study, Chang et al. reported an association between TLR1 and TLR6 in the TLR2-mediated activation of macrophages by HCV core and NS3 proteins [56]. While the involvement of TLR2 in sensing HCV core protein has been observed previously, infectious virions or enveloped HCV-like particles do not activate TLR2 [57]. However, another study showed that HCV core protein can activate TLR2 with decreased IL-6 production by human antigen-presenting cells after subsequent stimulation with TLR2 and TLR4 ligands [58]. HCV core protein may affect pDCs by reducing TLR9-triggered IFN-α as well as TNF-α and IL-10 production [59].

HCV core and NS3 antigens induce TLR1-, TLR2-, and TLR6-mediated inflammatory responses in corneal epithelial cells [60]. Compared to healthy controls, upregulation of TLR2 and TLR4 expression in peripheral monocytes was also observed in patients with chronic hepatitis C, with or without HIV coinfection [61]. An association between TLR4 signaling and the outcome of acute hepatitis C has also been reported [62]. Activation of the TLR3/TRIF signaling pathway by HCV NS5B, a viral RNA-dependent RNA polymerase, has been reported [63]. In addition to TLR3, HCV RNA can also be recognized by RIG-I, triggering the production of multiple cytokines, including type I IFN [42]. In a murine replicon model, the antiviral role of TLR4 activation in suppressing HCV replication was demonstrated [64]. An association between TLR4 gene polymorphisms and chronic HCV infection has also been reported in a Saudi Arabian population [65], suggesting a putative role of TLR4 in HCV infection. However, a larger genome-wide association study is required to validate these associations.

As HCV has an RNA genome, it is likely that TLR7 plays a role in the immune response against HCV infection. Both TLR3 and TLR7 have been suggested to coordinate protective immunity against HCV infection [45,66,67]. A decreased expression of TLR3 and TLR7 mRNA has been reported in chronic HCV patients with a decreased IFN-α expression compared to healthy controls [68,69,70]. An earlier study reported significantly elevated expression of TLR3 in individuals who spontaneously cleared the virus [71], suggesting a protective role of TLR3 in HCV genotype 3 infection. TLR7 and TLR8 were also found to be elevated in patients with liver cirrhosis [71]. A previous study reported that impaired TLR3- and TLR7/8-mediated cytokine responses may contribute to aggressive HCV recurrence after liver transplantation [72], also indicating the association of these molecules with HCV infection. However, more extensive in vivo studies are required to understand their use in protective immune responses against HCV infections.

It has been reported that TLR7 can sense HCV RNA in exosomes released from infected hepatocytes, inducing type I IFN response [73,74]. An in vitro study demonstrated that TLR7 can induce HCV immunity not only by IFN induction but also via an IFN-independent mechanism [75]. The antiviral roles of TLR7 and TLR8 have also been suggested in HCV infection [76]. In a previous study, it was shown that HCV could be recognized and inhibited by TLR7 and TLR8 via TNF-α production [77]. HCV genomic RNA-induced TLR7- and TLR8-mediated anti-HCV immune responses have been reported in various antigen-presenting cells [78]. Polymorphisms in TLR7 and/or TLR8 genes have been shown to modulate HCV infection outcomes [79,80,81], suggesting an association between these molecules and HCV infection. The TLR9 rs187084 C allele was reported to be associated with spontaneous virus clearance in women, suggesting the sex-specific effects of TLR9 promoter variants on spontaneous clearance in HCV infection and implying the role of the DNA sensor TLR9 in natural immunity against HCV infection [82]. In a recent study, an association between TLR9 gene polymorphisms and the outcome of the HCV-specific cell-mediated immune response was reported [83], indicating a putative role of TLR9 in HCV infection. A suitable small animal model is still lacking for HCV infection, which is essential for a proper understanding of the TLR response to HCV infection. Tree shrews appear to be a promising animal model for several important viral infections in humans, including hepatitis C virus [84], and show a higher degree of genetic similarity to primates than to rodents [85,86]. In an earlier study, we showed that HCV could trigger innate immune responses in the livers of chronically infected tree shrews, with significant induction of intrahepatic TLR3, TLR7, and TLR8 mRNA [87]. For simplicity, the findings obtained in different studies are shown in Figure 1, without indication of cell type/system, and highlight that various TLRs are implicated in HCV infection, which may influence viral pathogenesis. Therefore, a complete understanding of the TLR response in HCV infection is critical for designing new and successful therapeutic or preventive interventions.

## 3. Inhibition of Innate Immune Response by HCV Infection

The host has evolved multifaceted innate immune mechanisms to sense and counteract HCV infection. However, the success of innate IFN response in inhibiting HCV infection remains low. In a large proportion of patients, HCV persistence has been observed for decades despite the expression of hundreds of interferon-stimulated genes, indicating the inability of the IFN system to clear HCV infection [30,88,89]. In addition, HCV has developed multiple strategies for innate immune regulation, including proteolytic cleavage of molecules that play key roles in the induction of the IFN response, changes in IFN-induced effector proteins, interference with the function of CD8+ T cells, and immune escape in T- and B-cell epitopes [90]. HCV encodes several proteins, including core, NS3/4A, NS4B, and NS5A, which play active roles in inhibiting the innate immune response [42,91,92,93]. HCV frameshift (F) protein, which is expressed by a translational ribosomal frameshift [94], has also been suggested to play a role in the immune evasion mechanism [92].

NS3/4A plays a central role in HCV pathogenesis by cleaving several host proteins [95,96,97]. HCV NS3/4A has the potential to cleave TRIF and impair the TLR3-dependent signaling pathway [45,93,98]. It has also been reported that NS3/4A protease can disrupt RIG-I signaling by cleavage or delocalization of IFN-β promoter stimulator 1, also known as mitochondrial antiviral signaling protein (MAVS), preventing downstream activation of IRF-3 and IFN-β induction [93,98,99,100,101,102]. Notably, cleavage of MAVS by NS3/4A has been reported in the infected human liver, demonstrating the importance of this cleavage in HCV infection in vivo [101,103]. HCV proteins may interfere with IFN-induced intracellular signaling, which could be an important mechanism for viral persistence and treatment resistance [104]. HCV core may inhibit the IFN-signaling pathway by interfering with the Janus kinase/signal transducer and activator of the transcription pathway [105,106,107]. Several studies have reported the inhibition or degradation of STAT1 by HCV core protein [104,108,109,110,111], which may inhibit the JAK/STAT-signaling pathway of the host response. Reduced levels of STAT2 phosphorylation caused by HCV core proteins have also been reported [112,113].

Several studies have shown the implication of HCV core protein in the activation of the NF-κB pathway for inducing inflammatory response [114,115,116,117]; however, an implication of HCV core protein in the suppression of the NF-κB pathway has also been reported [118]. STAT3 has been found to be downregulated in HCV-infected livers and in Huh7 cells [119], which may favor viral replication. Impaired TLR4 signaling in HCV-infected dendritic cells has been previously demonstrated [120]. It has been reported that NS4B can interact with the stimulator of IFN genes (STING), which may cause inhibition of downstream signaling [91,121,122]. HCV NS4A, NS4B, and NS5A may inhibit IFN-β induction, contributing to the persistence of this virus [63]. NS5A can directly bind to MyD88, a major adaptor molecule in TLR signaling, inhibiting the activation of TLR-mediated cytokine production [123]. NS5A can induce IL-8 expression, associated with the interruption of IFN-α [124]. NS5A also blocks the antiviral activity of 2′–5′ oligoadenylate synthetase (2′–5′ OAS) [125]. It has been reported that HCV may utilize NS5A and E2 to inhibit PKR-mediated antiviral defense [126,127,128]. A previous study also suggested a role for NS5A in the inhibition of the IFN response that is activated by HCV via RIG-I and MDA5 [129]. NS5A may interact with nucleosome assembly protein 1-like 1 (NAP1L1), a nuclear-cytoplasmic chaperone, which may downregulate genes essential for innate immunity, such as RIG-I- and TLR3-mediated responses [130].

Chronic HCV infection results in impaired TLR response in pDCs as well as impaired activation of naive CD4 T cells, with reduced activation marker (HLA-DR) and cytokine (IFN-α) expression upon R-848 stimulation [131]. Samrat et al. reported that HCV core and F protein can induce poor T-cell responses, resulting in low granzyme B expression by CD4+ and CD8+ T cells [92]. It has been reported that HCV p7 protein has an immune evasion function, which may suppress antiviral IFN function by counteracting IFN-inducible protein 6-16 (IFI6-16) [132]. Based on these findings, it is assumed that HCV and its proteins play a crucial role in inhibiting or suppressing the host innate immune response (Figure 2) by different known and unknown mechanisms; therefore, a clear understanding of immune inhibition or evasion is critical for devising new therapeutic and preventive strategies to control HCV infection.

## 4. Potential of TLR Agonists as Immunomodulators

A large number of viruses have been shown to trigger innate immunity via TLRs, which have been found important in the outcome of many viral infections [133,134,135], suggesting manipulating the TLR response could serve as a therapeutic avenue against viral infections. Although there is a great therapeutic success against HCV infection with newly approved drugs, direct-acting antivirals (DAAs) [136], cirrhosis and HCC have been reported in patients following viral clearance [137]. Moreover, current HCV treatment approaches are not effective in preventing recurrent infections. Notably, high treatment cost has restricted its use in economically weak countries. Therefore, there is an urgent need to develop an alternative therapeutic approach as well as an effective, safe, and affordable vaccine against HCV. There is a growing interest in the use of TLR agonists as vaccine adjuvants [138], which are capable of stimulating innate and adaptive immune responses, thereby improving vaccine efficacy. Recently, TLR agonists have received much attention as immunomodulators with the ability to induce the production of IFN, proinflammatory cytokines, and chemokines and have been found to be promising against many viral infections, including hepatitis B virus and human immunodeficiency virus type 1 (HIV-1) [35,139,140,141]. Manipulating TLR response has also been found to be promising against SARS-CoV-2 infection. In a ferret model, it has been shown that the injection of TLR2/6 agonist INNA-051 significantly reduced SARS-CoV-2 viral RNA levels in the nose and throat [142]. Additionally, it has been proposed that TLR agonists, including imiquimod, an immune stimulator of TLR7, could serve as an effective therapeutic approach in the early stages of COVID-19 [143].

Chronic HCV infection induces weak cellular immune responses against viral antigens, and viral clearance after acute hepatitis or therapy requires strong and multispecific antiviral CD4+ and CD8+ T-cell responses [144,145]. TLR agonists may play an important role in enhancing immunotherapeutic effects [33,34,35,138,146]. The use of TLR agonists as immunomodulators to enhance the immune response in chronic HCV infection could be of great interest for the control of HCV infection. Isatoribine, an agonist of TLR7, showed dose-dependent changes in immunologic biomarkers and antiviral effects against HCV infection [147]. The antiviral activity of the synthetic TLR7 agonist was shown to be associated with the stimulation of antiviral genes, such as IRF7, but not with the activation of the IFN-responsive STAT-1 transcription factor [75]. A previous study demonstrated that oral administration of resiquimod, a TLR 7/8 agonist, transiently reduced viral levels but was associated with adverse effects similar to IFN-α [148]. Another study showed that TLR3/4/7/8/9 agonists could induce anti-HCV activity in PBMC supernatants, correlating with IFN-α and the IFN-induced antiviral biomarker 2′,5′-oligoadenylate synthase induction. However, TLR4 and TLR8 agonists induce the proinflammatory cytokines IL-1β and TNF-α at concentrations similar to those inducing antiviral activity, raising concerns regarding adverse side effects [149]. In a randomized clinical trial, oral administration of ANA773, a prodrug of the TLR7 agonist, resulted in an IFN-dependent response leading to a significant decrease in serum HCV RNA levels, with mild to moderate adverse events [150].

It has been reported that TLR7/9 agonists may enhance the inhibition of infectivity and IFN-α production by pDCs, suggesting pDCs could serve as a drug target against HCV infection [151]. It also suggests the possibility of using TLR7/9 agonists in HCV vaccine development. An earlier study showed that although a hepatitis C viral-like particle vaccine adjuvanted with TLR2 agonists, R_4_Pam_2_Cys and E_8_Pam_2_Cys, induced higher antibody titers in mice, it did not induce stronger NAb responses compared to vaccines without adjuvants [152], suggesting the usefulness of TLR agonists as vaccine adjuvants for HCV vaccines. Using appropriate adjuvants in HCV vaccine candidates, the induction of a strong T- and B-cell immune response may be enhanced [153]. In an in vitro study with an HCV-infected hepatoma cell line, Huh7.5, Dominguez-Molina et al. showed that TLR agonists can enhance antiviral pDCs function primarily through IFN-α production against HCV infection [151].

## 5. Conclusions

From the currently available data, it is understood that there is a differential expression of TLRs in HCV infection. Moreover, TLR3 and TLR7 may play a protective role against HCV infection. However, the exact role of the TLR response to HCV infection requires further extensive study, which also requires a suitable small animal model. HCV has developed multiple strategies to inhibit the innate immune response toward establishing a chronic infection, which also requires additional investigation for developing new therapeutic approaches. As there are mixed effects, an extensive study is still required to select the best-suited TLR agonist for use as a vaccine adjuvant for HCV vaccine candidates. Overall, a clear understanding of TLR interactions in HCV infection is critical for providing new therapeutic and preventive approaches to fight the disease, including TLR agonist-adjuvanted HCV vaccines.

## Figures and Tables

**Figure 1 ijms-23-05475-f001:**
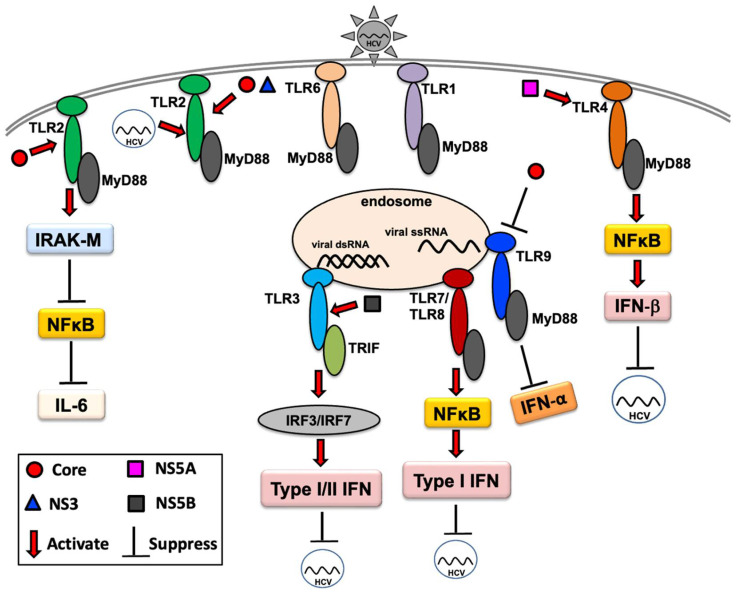
TLR response to HCV infection. Red arrows indicate the induction/activation of components of TLR signaling by HCV or its proteins; black lines indicate the inhibition of the host innate immune response or inhibition of HCV replication, as appropriate.

**Figure 2 ijms-23-05475-f002:**
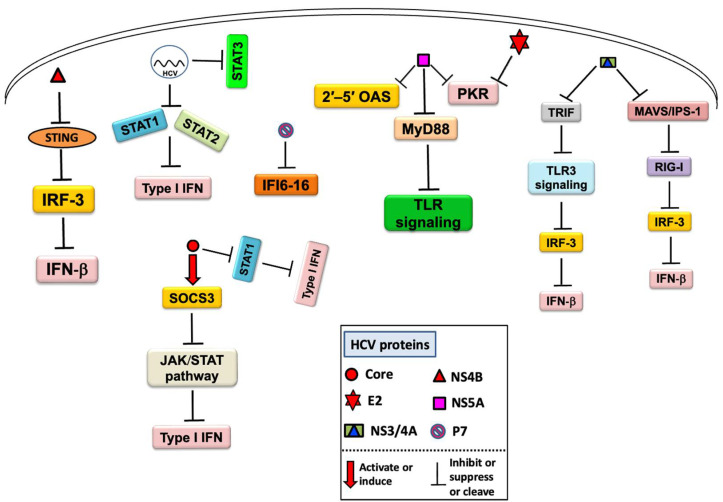
An overview of the mechanism of the host innate immune response inhibition by HCV and its proteins.

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
