# Peer review of "Toll-like Receptor Response to Hepatitis C Virus Infection: A Recent Overview"

_ijms, 2022, doi:10.3390/ijms23105475_

Round 1

Reviewer 1 Report

Summary:

The authors have written a review focusing on the interplay between HCV infection and the TLR response part of the innate immunity.The review is logically composed of 4 parts. First the authors present the TLR response and HCV infection. Secondly, they present the mechanism involved in the recognition of HCV infection by the different Toll-Like Receptors. The authors then present the escape mechanism put in place by the virus to evade TLR signalling. Finally, in their 4th section, the authors aim to introduce therapeutic avenue to take advantage of the TLR response to treat HCV infection.

Comments:

In my opinion the review is overall well-written and treat the subject rather well. In addition, the English is impeccable. However, I have some minor comments to improve the review:

  • The correlation between the TLR response and HCV chronic infection in patients is minimally presented in the review and could be further expanded.

  • The 4th section "Potential of TLR agonists as immunomodulators" should be further expanded. In particular I would add a couple of paragraphs introducing similar attempts at manipulating the TLR response as therapeutic avenue against other viruses. It would help to open the review and present to the field how this method can be successful against other pathogens.

Author Response

In my opinion the review is overall well-written and treat the subject rather well. In addition, the English is impeccable. However, I have some minor comments to improve the review:

Response: We are very grateful to the reviewer for his/her sincere comments.

  • The correlation between the TLR response and HCV chronic infection in patients is minimally presented in the review and could be further expanded.

Response: According to the reviewer comments, we have further updated the manuscript by adding the information of the correlation between the TLR response and HCV chronic infection in patients (page 3, line 99-103; 141-143).

  • The 4th section "Potential of TLR agonists as immunomodulators" should be further expanded. In particular I would add a couple of paragraphs introducing similar attempts at manipulating the TLR response as therapeutic avenue against other viruses. It would help to open the review and present to the field how this method can be successful against other pathogens.

Response: In line with reviewer comments, we have updated the text (page 7, line 244-246; 253-264).

Reviewer 2 Report

In this review, Kayesh et al discussed the importance of toll like receptors (TLRs) in Hepatitis C viral infection (HCV). According to their analysis, there is a differential expression of TLRs in HCV infection. However, the authors suggest that there should be further extensive studies to conclude the exact role of the TLR response to HCV infection.

Specific comments:

  1. Hepatitis C can be treated with therapeutics that stop the virus multiplying inside the body. Therefore, the authors need to emphasise the reason for looking at the importance of toll like receptors (TLRs) in Hepatitis C viral infection and new therapeutic approaches against HCV infection.

Author Response

Specific comments:

  1. Hepatitis C can be treated with therapeutics that stop the virus multiplying inside the body. Therefore, the authors need to emphasise the reason for looking at the importance of toll like receptors (TLRs) in Hepatitis C viral infection and new therapeutic approaches against HCV infection.

Response: In response to reviewer comments, we have updated the text including the importance of TLRs in HCV infection and new therapeutic approaches against HCV infection (page, line 244-255).

Round 2

Reviewer 1 Report

Authors addressed my comments and improved the manuscript accordingly. Manuscript is now ready for publication.